# Punch Incision versus Elliptical Excision for Epidermal Inclusion Cysts: Systematic Review and Meta-Analysis

**Kengo Mukuda** [1],* and **Jun Watanabe** [2,3]

1   Division of Infectious Diseases, Faculty of Medicine, Tottori University, Yonago-City 683-8504, Japan
2   Department of Surgery, Division of Gastroenterological, General and Transplant Surgery,
    Jichi Medical University, Shimotsuke-City 329-0498, Japan; m06105jw@jichi.ac.jp
3   Division of Community and Family Medicine, Jichi Medical University, Shimotsuke-City 329-0498, Japan
*   Correspondence: mukudak@tottori-u.ac.jp; Tel.: +81-859-38-6076

**Abstract:** Punch incision is an alternative to elliptical excision for treating epidermal inclusion cysts, but its efficacy has not been systematically reviewed. This study assessed the efficacy and safety of punch incision versus elliptical excision for epidermal inclusion cysts. Randomized controlled trials published through January 2021 that evaluated the performance of punch incision versus elliptical excision on epidermal inclusion cysts were identified through electronic databases and clinical registries. Version 2 of the Cochrane risk-of-bias tool for randomized trials tool was used. Review Manager software was used for the meta-analysis. Two trials (100 participants) were identified. The primary outcomes were recurrence rate (risk ratio, 2.40; 95% confidence interval [CI], 0.37–15.60 [favoring elliptical excision]), mean operative time (mean difference [MD], −5.28; 95% CI, −12.72 to 2.16 [favoring punch incision]), and mean postoperative wound length (MD, −11.67; 95% CI, −20.59 to −2.76 [favoring punch incision]). The evidence was low to moderate due to the small sample size and its considerable heterogeneity. The use of punch incision shortened the mean postoperative wound length and had comparable safety to that of elliptical excision.

**Keywords:** epidermal inclusion cyst; epidermoid cyst; meta-analysis; pilar cyst; punch incision; randomized controlled trial



## 1. Introduction

Epidermal inclusion cysts (EICs) are common cysts that are 0.3–9 cm in diameter, contain keratin, and are undercoated by the true epidermis [1]. In a university hospital setting, there were approximately 1000 diagnoses of epidermal cysts over six years [2]. Patients with EICs often require cyst removal due to increasing size, poor appearance, foul-smelling discharge, or pain. Although elliptical excision and manual blunt dissection of the cyst wall is most commonly used to achieve complete excision for cleaning up such lesions [3], the procedure often results in a longer scar than certain less invasive techniques [4].

Punch incision is an alternative to elliptical excision for the removal of EICs. Punch incision features a shorter scar and shorter healing time, but the efficacy of the two procedures has never been systematically reviewed and analyzed.

This systematic review and meta-analysis of RCTs aimed to assess the efficacy and safety of punch incision versus elliptical excision for EICs.

## 2. Methods

This systematic review was performed according to the Preferred Reporting Items for Systematic Reviews and Meta-analyses 2020 statement [5] (Appendices A and B). We registered our protocol at protocol.io (https://www.protocols.io/view/punch-incision-versus-elliptical-excision-for-epid-brgvm3w6, accessed on 15 January 2021). There were

not any amendments to information provided at registration or in the protocol. None of the funders played any role in the study or decision to publish our findings.

### 2.1. Type of Studies

We included individual and cluster randomized controlled trials (RCTs) that assessed punch incision versus elliptical excision of EICs. We did not apply language or country restrictions. We included all literature, including published articles, unpublished articles, abstracts of conferences, and letters. We excluded non-RCTs and crossover trials. We did not exclude studies based on the observation period. The inclusion criteria were non-infected EICs that were clinically diagnosed by physicians. Cases of infected EICs were excluded.

We included patients with non-infected EICs that were clinically diagnosed by physicians.

The intervention was a punch incision using a dermal punch biopsy trephine to make a small, round incision. The comparator was an elliptical excision using a standardized procedure.

### 2.2. Type of Outcomes

The primary outcomes were: (1) recurrence rate, (2) mean operative time, and (3) mean postoperative wound length. The recurrence rate was defined as the reported recurrence by participants during a visit or telephone interview within the 12-month follow-up period. The mean operative time was defined as the procedural time recorded from the time of the first incision to the time of suture closure. The mean postoperative wound length was defined as the length measured at the time of wound closure. A secondary outcome was set as all adverse events that were determined by the original authors. All adverse events were recorded during follow-up.

### 2.3. Search Method

We searched the following electronic databases on January 12, 2021: (1) the Cochrane Central Register of Controlled Trials (CENTRAL); (2) MEDLINE via PubMed (1966 to present); and (3) EMBASE via PROQUEST (1988 to present) (Appendix C). We also searched the following databases on January 12, 2021 for ongoing or recently completed trials: (1) the World Health Organization International Clinical Trials Platform Search Portal (ICTRP) and (2) ClinicalTrials.gov (Appendix D). We sorted the reference lists of each study, including international guidelines [6], as well as the reference lists of eligible studies and articles citing eligible studies. We requested unpublished or additional data from the authors of the original studies. Our protocol was registered at protocol.io (https://www.protocols.io/view/punch-incision-versus-elliptical-excision-for-epid-brgvm3w6, accessed on 15 January 2021).

### 2.4. Data Collection

Two reviewers (KM and JW) independently screened the titles and abstracts of the articles. The articles that met the inclusion criteria were subjected to full-text review. Each reviewer independently performed the study selection. We contacted the original authors if there were any disagreements regarding the articles. Two reviewers compared the lists. Any differences in opinion were resolved by discussion.

### 2.5. Data Extraction

Two reviewers (KM and JW) independently extracted data from the included trials using the data collection form. We abstracted general information including study design, author names, publication year, country of origin, and baseline characteristics (sample size, mean age, and sex distribution), as well as the primary and secondary outcomes of interest. Any disagreements were resolved through discussion.

For continuous data, we reported the results of the individual outcomes as mean and standard deviation (SD). Because SD was not reported in Cheeley's RCT [7], we imputed SD for Lee's RCT [4] by invoking the previous research [8].

*2.6. Analysis*

Two reviewers (KM and JW) separately assessed the risk of bias. We used Version 2 of the Cochrane risk-of-bias tool for randomized trials [9]. The tool has the following five domains: (1) bias arising from the randomization process; (2) bias due to deviations from the intended interventions; (3) bias due to missing outcome data; (4) bias in measurement of the outcome; and (5) bias in selection of the reported result. Any disagreements were resolved through discussion.

We conducted author inquiries up to two times by email to obtain the relevant data.

We performed a meta-analysis and calculated the relative risk (RR) and 95% confidence interval (CI) for the binary outcome of recurrence rate. We performed a meta-analysis and calculated the mean difference (MD) and 95% CI for the continuous variables of mean operative time and mean postoperative wound length. We summarized adverse events according to the definition provided by each original article, but we did not perform a meta-analysis. We used a forest plot to show the results of the meta-analysis.

Statistical heterogeneity was evaluated by visual inspection of the forest plots and calculation of the I2 statistic (I2 values of 0–40%: might not be important; 30–60%: may represent moderate heterogeneity; 50–90%: may represent substantial heterogeneity; and 75–100%: indicates considerable heterogeneity) [9]. In cases of substantial heterogeneity (I2 > 50%), we considered the reason for the heterogeneity in the primary outcomes. The Cochrane $\chi^2$ test (Q-test) was performed to determine the I2 statistic, and *p* values less than 0.10 were defined as statistically significant.

We searched the clinical trial registry systems (ClinicalTrials.gov and ICTRP) and performed an extensive literature search for unpublished trials. We did not create a funnel plot or perform the Egger test because we found fewer than 10 trials [9].

A meta-analysis was performed with Review Manager software (RevMan 5.4.1., The Cochrane Collaboration, Copenhagen, Denmark) using a random effects model.

We did not conduct subgroup and sensitivity analyses due to a lack of sufficient data.

A summary of findings (SoF) table was created for the following outcome based on the Cochrane Handbook [9]: (1) recurrence rate; (2) mean operative time; (3) mean postoperative wound length; and (4) all adverse events. The point estimate of effect, 95% CI, and certainty of evidence are listed for each outcome in the SoF table (Table 1).

**Table 1.** Summary of the characteristics of the eligibility studies.

| Authors [Ref Number] | Year | Subject Number | Age (Years) | Procedure | Female (%) | Follow-Up (Months) | Cyst Size (mm) |
|---|---|---|---|---|---|---|---|
| [7] | 2018 | 40 | 59.6 | Punch incision | 1 | 16 | 19.1 |
|  |  |  |  | Elliptical excision | 10 |  | 15.5 |
| [4] | 2006 | 60 | 37.5 | Punch Technique | 32 | 14–29 | 1.14 |
|  |  |  |  | Excision Technique | 45 |  | 1.06 |

We included grading to evaluate the quality of evidence-based recommendations on the Grading of Recommendations Assessment, Development, and Evaluation approach for each SoF table [9].

After removing duplicates, we identified 54 records during the search conducted in January 2021. Two trials were included in the qualitative synthesis. Finally, we included 100 participants from two trials in the quantitative synthesis.

## 3. Results

### 3.1. Search Results

Figure 1 shows the flowchart of the study selection process. We found 54 records in the initial screening on 12 January 2021. After the full-text screening, we included two trials, which included 100 participants and compared punch incision with elliptical excision in the qualitative synthesis [4,7]. Table 1 summarizes the characteristics of the included studies. One RCT was performed in the USA, while the other was performed in Taiwan. The mean subject ages were 60 and 38 years, while the sample sizes were 40 and 60, respectively. In the former study, most of the included subjects were men.

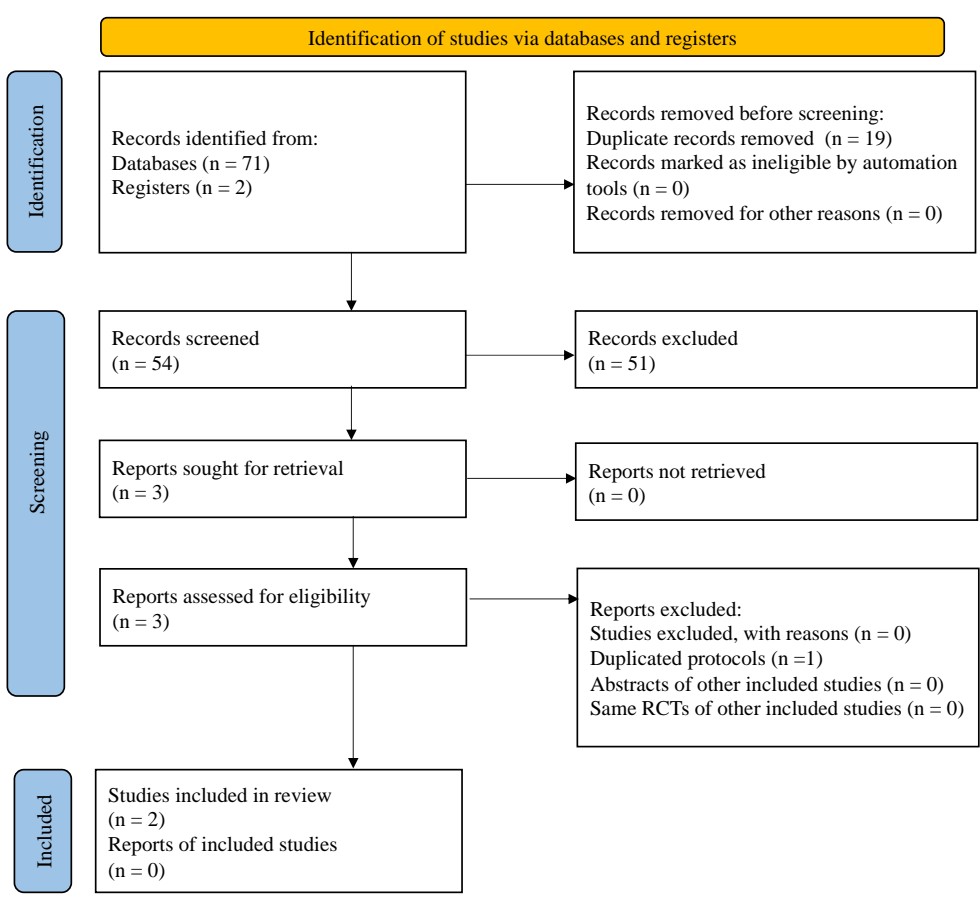

**Figure 1.** Flow diagram of the literature search results.

The risk of bias for each study is shown in Table 2 (and Appendix E). Overall, we conducted two trials of the primary outcomes in the meta-analysis.

**Table 2.** Quality scores for the eligibility studies for recurrence rate.

| Authors | Risk of Bias 2 Tool Assessment | | | | | |
|---|---|---|---|---|---|---|
| [Ref Number] | Bias Arising from the Randomization Process | Bias Due to Deviations from Intended Interventions | Bias Due to Missing Outcome Data | Bias in Measurement of the Outcome | Bias in Selection of the Reported Results | Overall Risk of Bias |
| [7] | Some concerns | Low | Low | Low | Low | Low |
| [4] | High | Low | Low | Low | Some concerns | High |

### 3.2. Meta-Analysis

#### 3.2.1. Recurrence Rate

Two studies reported recurrence rates. The use of punch incision may make little to no difference in outcomes (two studies, 100 participants) as follows: RR, 2.40; 95% CI, 0.37–15.60; I2 = 0%; low certainty of evidence (Figure 2).

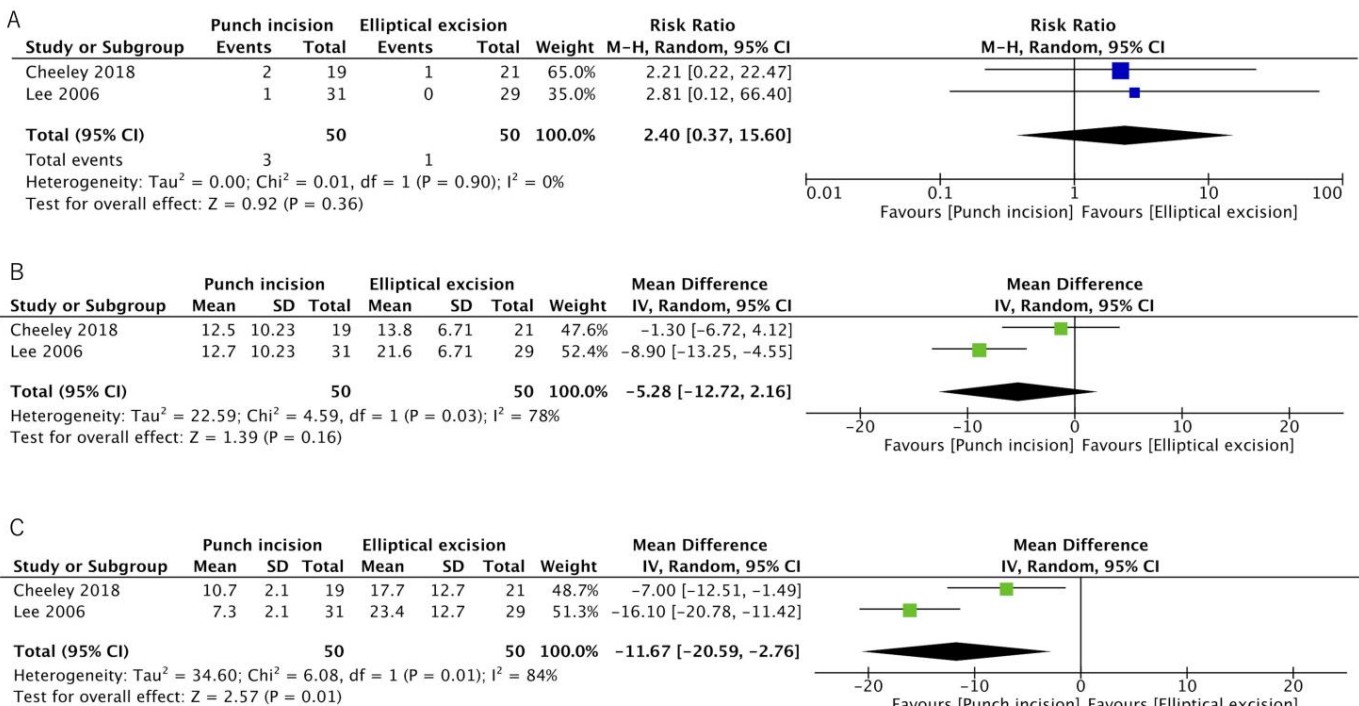

**Figure 2.** Forest plots. (**A**) recurrence rate, (**B**) mean operative time, and (**C**) mean postoperative wound length.

#### 3.2.2. Mean Operative Time

The two studies reported operative times (in minutes). The use of punch incision may result in a slight reduction in outcomes (2 studies, 100 participants) as follows: MD, −5.28; 95% CI, −12.72 to 2.16; I2 = 78%; low certainty of evidence (Figure 2).

#### 3.2.3. Mean Postoperative Wound Length

The two studies reported the postoperative wound lengths (in millimeters). The use of punch incision likely resulted in a reduction in outcome (2 studies, 100 participants) as follows: MD −11.67; 95% CI, −20.59 to −2.76; I2 = 84%; moderate certainty of evidence (Figure 2).

#### 3.2.4. All Adverse Events

Among the included studies, only one reported adverse events [7]. Both groups had one or two cases of dehiscence, infection, bleeding, tenderness, and drainage. The punch incision group had one case of bleeding and two cases of drainage, while the elliptical excision group had one case of hematoma (Table 3).

**Table 3.** Summary of Findings (SoF).

| Outcomes | Anticipated Absolute Effects (95% CI) * | | Relative Effect (95% CI) | Patient Number (Studies) | Certainty | Comments |
| | Risk with Elliptical Excision | Risk with Punch Incision | | | | |
|---|---|---|---|---|---|---|
| Recurrence rate | 2% | 5% (0.7 to 27.2) | RR 2.46 (0.35 to 13.59) | 100 (2 RCTs) | LOW a | Punch incision may make little to no difference in recurrence rate. |
| Mean operative time | - | MD 5.28 min lower [−12.72 to 2.16] | - | 100 (2 RCTs) | LOW a,b | Punch incision may result in a slight reduction in operative time. |
| Mean length of the postoperative wound | - | MD 11.67 mm lower [−20.59 to −2.76] | - | 100 (2 RCTs) | MODERATE a | Punch incision likely has a shorter operative time. |
| All adverse events | Just one study reported the adverse events. | | | 100 (2 RCTs) | MODERATE a | Both had one or two of Dehiscence, Infection, Bleeding, Tenderness, and Drainage. Punch incision had Bleeding and Drainage. Elliptical excision had a hematoma. |

Punch Incision Compared to Elliptical Excision for Epidermal Inclusion Cysts. Patient or Population: Epidermal Inclusion Cysts, Intervention: Punch Incision, Comparison: Elliptical Excision CI, confidence interval; RR, risk ratio; MD, mean difference. * The risk in the intervention group (and its 95% CI) is based on the assumed risk in the comparison group and the relative effect of the intervention (and its 95% CI). GRADE Working Group grades of evidence; High certainty: We are very confident that the true effect lies close to that of the estimated effect. Moderate certainty: We are moderately confident in the estimated effect. The true effect is likely to be close to the estimated effect, but there is a possibility that it is substantially different. Low certainty: our confidence in the estimated effect is limited; the true effect may be substantially different from the estimated effect. Very low certainty: we have very little confidence in the estimated. a. Downgraded because of imprecision due to the small sample size. b. Downgraded because of inconsistency that there was represent considerable heterogeneity.

## 4. Discussion

The present study meta-analyzed the differences between punch incision and elliptical excision in EICs and showed that punch incision shortened the mean postoperative wound length and had comparable safety. Nevertheless, we should acknowledge the very small number of articles included in this review, even though the results from the integrated RCTs on punch incision for including epidermal cysts may be useful for evidence-based practice.

Punch incision may result in little to no difference in recurrence rate. However, attention should be paid to the tendency of a higher recurrence rate in punch incision. Previous studies reported recurrence rates of 3–8%, and the recurrence rates of the back and ear were higher than those of other body parts [10]. In Lee's RCT, the only recurrence encountered was a cyst larger than 2 cm located in the postauricular area [4]. For cysts of the back and ear or those larger than 2 cm, the use of punch incision might be better for preventing recurrence.

Use of the punch incision may result in a slight reduction in mean operative time. However, this difference may not be detectable because of the short operative time. The use of punch incision may increase the operative time when cysts are larger than 2 cm, but we could not conduct a subgroup analysis because of the small number of studies.

A decrease of 11.67 mm in the mean postoperative wound length might be insignificant for physicians but important for patients. Postoperative scar cosmesis is very important for women, especially young women, and scars can affect patients psychologically [11–13]. Moreover, postoperative scar length may be more important than scar aspects including width, thickness, color, shape, and the presence of suture tracks [14]. However, there are several scar assessment scales. Of them, the Vancouver Scar Scale was the most widely used, while the Patient and Observer Scar Assessment Scale was the most comprehensive [12]. It is better to include these scales while conducting further studies to comprehensively assess scars and patient satisfaction.

We found no life-threatening adverse events in either group. However, despite the small number of patients, hematoma, bleeding, and infection were important adverse events. In a practical setting, physicians can minimize bleeding with the co-administration of adrenaline and a local anesthetic or compression [15,16]. Antibiotics are used to prevent infection, especially for patients with any risk factors such as underlying immuno-

suppressed state, diabetes mellitus, use of nonsterile biopsy instruments or sutures, or inadequate or no postoperative antibiotic coverage [16].

The present study has several limitations. First, only two studies were included. The small number of included studies may make it difficult to evaluate the true efficacy of punch incision. However, the present study might be important because of the lack of previous review. Second, the quality of the included studies was downgraded because of considerable heterogeneity. Further investigations are needed to overcome the unequal sex distribution and incomplete randomization.

In conclusion, we demonstrated that punch incision shortened the mean postoperative wound length and had comparable safety to that of elliptical excision for EICs. These findings suggest that physicians should consider using punch biopsy in cases in which patient preference and cyst characteristics are amenable to it. Further large and well-designed RCTs are needed to verify our findings and unravel the risk factors that would impact the efficacy and safety of punch incision.

**Author Contributions:** Conceptualization, K.M. and J.W.; methodology, K.M. and J.W.; software, K.M. and J.W.; validation, K.M. and J.W.; formal analysis, K.M. and J.W.; investigation, K.M. and J.W.; resources, N/A; data curation, K.M. and J.W.; writing—original draft preparation, K.M.; writing—review and editing, K.M. and J.W.; visualization, K.M. and J.W.; supervision, J.W.; project administration, K.M. and J.W.; funding acquisition, N/A. All authors have read and agreed to the published version of the manuscript.

**Funding:** This research received no external funding.

**Institutional Review Board Statement:** Not applicable.

**Informed Consent Statement:** Not applicable.

**Conflicts of Interest:** The authors declare no conflict of interest.

## Appendix A. PRISMA 2020 Abstract Checklist

| Section and Topic | Item | Checklist Item | Reported (Yes/No) |
|---|---|---|---|
| | | TITLE | |
| Title | 1 | Identify the report as a systematic review. | Yes |
| | | BACKGROUND | |
| Objectives | 2 | Provide an explicit statement of the main objective(s) or question(s) the review addresses. | Yes |
| | | METHODS | |
| Eligibility criteria | 3 | Specify the inclusion and exclusion criteria for the review. | Yes |
| Information sources | 4 | Specify the information sources (e.g., databases, registers) used to identify studies and the date when each was last searched. | No written in main text |
| Risk of bias | 5 | Specify the methods used to assess risk of bias in the included studies. | Yes |
| Synthesis of results | 6 | Specify the methods used to present and synthesise results. | Yes |
| | | RESULTS | |
| Included studies | 7 | Give the total number of included studies and participants and summarise relevant characteristics of studies. | Yes |
| Synthesis of results | 8 | Present results for main outcomes, preferably indicating the number of included studies and participants for each. If meta-analysis was done, report the summary estimate and confidence/credible interval. If comparing groups, indicate the direction of the effect (i.e., which group is favoured). | Yes |
| | | DISCUSSION | |
| Limitations of evidence | 9 | Provide a brief summary of the limitations of the evidence included in the review (e.g., study risk of bias, inconsistency and imprecision). | Yes |
| Interpretation | 10 | Provide a general interpretation of the results and important implications. | Yes |
| | | OTHER | |
| Funding | 11 | Specify the primary source of funding for the review. | No written in main text |
| Registration | 12 | Provide the register name and registration number. | No written in main text |

## Appendix B. PRISMA2020 Main Checklist

| Topic | No. | Item | Location Where Item Is Reported |
|---|---|---|---|
| TITLE | | | |
| Title | 1 | Identify the report as a systematic review. | Line 1–3 |
| ABSTRACT | | | |
| Abstract | 2 | See the PRISMA 2020 for Abstracts checklist | |
| INTRODUCTION | | | |
| Rationale | 3 | Describe the rationale for the review in the context of existing knowledge. | Line 37–39 |
| Objectives | 4 | Provide an explicit statement of the objective(s) or question(s) the review addresses. | Line 40–41 |
| METHODS | | | |
| Eligibility criteria | 5 | Specify the inclusion and exclusion criteria for the review and how studies were grouped for the syntheses. | Line 49–61 |
| Information sources | 6 | Specify all databases, registers, websites, organisations, reference lists and other sources searched or consulted to identify studies. Specify the date when each source was last searched or consulted. | Line 71–81 |
| Search strategy | 7 | Present the full search strategies for all databases, registers and websites, including any filters and limits used. | Appendices C and D |
| Selection process | 8 | Specify the methods used to decide whether a study met the inclusion criteria of the review, including how many reviewers screened each record and each report retrieved, whether they worked independently, and if applicable, details of automation tools used in the process. | Line 82–87 |
| Data collection process | 9 | Specify the methods used to collect data from reports, including how many reviewers collected data from each report, whether they worked independently, any processes for obtaining or confirming data from study investigators, and if applicable, details of automation tools used in the process. | Line 88–96 |
| Data items | 10a | List and define all outcomes for which data were sought. Specify whether all results that were compatible with each outcome domain in each study were sought (e.g., for all measures, time points, analyses), and if not, the methods used to decide which results to collect. | Line 62–70 |
| | 10b | List and define all other variables for which data were sought (e.g., participant and intervention characteristics, funding sources). Describe any assumptions made about any missing or unclear information. | Line 88–96 Table 1 |
| Study risk of bias assessment | 11 | Specify the methods used to assess risk of bias in the included studies, including details of the tool(s) used, how many reviewers assessed each study and whether they worked independently, and if applicable, details of automation tools used in the process. | Line 97–104 |
| Effect measures | 12 | Specify for each outcome the effect measure(s) (e.g., risk ratio, mean difference) used in the synthesis or presentation of results. | Line 105–110 |

| Topic | No. | Item | Location Where Item Is Reported |
|---|---|---|---|
| Synthesis methods | 13a | Describe the processes used to decide which studies were eligible for each synthesis (e.g., tabulating the study intervention characteristics and comparing against the planned groups for each synthesis (item 5)). | Table 1 |
| | 13b | Describe any methods required to prepare the data for presentation or synthesis, such as handling of missing summary statistics, or data conversions. | Line 94–96 |
| | 13c | Describe any methods used to tabulate or visually display results of individual studies and syntheses. | Line 128–130 |
| | 13d | Describe any methods used to synthesize results and provide a rationale for the choice(s). If meta-analysis was performed, describe the model(s), method(s) to identify the presence and extent of statistical heterogeneity, and software package(s) used. | Line 121–127 |
| | 13e | Describe any methods used to explore possible causes of heterogeneity among study results (e.g., subgroup analysis, meta-regression). | Line 123 |
| | 13f | Describe any sensitivity analyses conducted to assess robustness of the synthesized results. | Line 123 |
| Reporting bias assessment | 14 | Describe any methods used to assess risk of bias due to missing results in a synthesis (arising from reporting biases). | Line 97–103 |
| Certainty assessment | 15 | Describe any methods used to assess certainty (or confidence) in the body of evidence for an outcome. | Line 130–132 |
| RESULTS | | | |
| Study selection | 16a | Describe the results of the search and selection process, from the number of records identified in the search to the number of studies included in the review, ideally using a flow diagram. | Line 138–141 |
| | 16b | Cite studies that might appear to meet the inclusion criteria, but which were excluded, and explain why they were excluded. | Figure 1 |
| Study characteristics | 17 | Cite each included study and present its characteristics. | Line 141–144 Table 1 |
| Risk of bias in studies | 18 | Present assessments of risk of bias for each included study. | Line 143–144 Table 2, Appendix E |
| Results of individual studies | 19 | For all outcomes, present, for each study: (a) summary statistics for each group (where appropriate) and (b) an effect estimate and its precision (e.g., confidence/credible interval), ideally using structured tables or plots. | Line 154–180 Figure 2 |
| Results of syntheses | 20a | For each synthesis, briefly summarise the characteristics and risk of bias among contributing studies. | Line 139–144 Tables 1 and 2 Appendix E |
| | 20b | Present results of all statistical syntheses conducted. If meta-analysis was done, present for each the summary estimate and its precision (e.g., confidence/credible interval) and measures of statistical heterogeneity. If comparing groups, describe the direction of the effect. | Line 154–180 Table 3 Figure 2 |

| Topic | No. | Item | Location Where Item Is Reported |
|---|---|---|---|
| | 20c | Present results of all investigations of possible causes of heterogeneity among study results. | Line 111–117 |
| | 20d | Present results of all sensitivity analyses conducted to assess the robustness of the synthesized results. | Line 123 |
| Reporting biases | 21 | Present assessments of risk of bias due to missing results (arising from reporting biases) for each synthesis assessed. | Line 148–149 Table 2 Appendix E |
| Certainty of evidence | 22 | Present assessments of certainty (or confidence) in the body of evidence for each outcome assessed. | Table 3 |
| DISCUSSION | | | |
| Discussion | 23a | Provide a general interpretation of the results in the context of other evidence. | Line 198–203 |
| | 23b | Discuss any limitations of the evidence included in the review. | Line 234–235 |
| | 23c | Discuss any limitations of the review processes used. | Line 231–234 |
| | 23d | Discuss implications of the results for practice, policy, and future research. | Line 235–236 |
| OTHER INFORMATION | | | |
| Registration and protocol | 24a | Provide registration information for the review, including register name and registration number, or state that the review was not registered. | Line 44–45 |
| | 24b | Indicate where the review protocol can be accessed, or state that a protocol was not prepared. | Line 44–45 |
| | 24c | Describe and explain any amendments to information provided at registration or in the protocol. | Line 45–46 |
| Support | 25 | Describe sources of financial or non-financial support for the review, and the role of the funders or sponsors in the review. | Line 46–47 |
| Competing interests | 26 | Declare any competing interests of review authors. | Line 47–48 |
| Availability of data, code and other materials | 27 | Report which of the following are publicly available and where they can be found: template data collection forms; data extracted from included studies; data used for all analyses; analytic code; any other materials used in the review. | Line 72–79 Appendices C and D |

## Appendix C. The Electronic Database Search Strategy

CENTRAL search strategy

([mh "Epidermal Cyst"] OR ((Epidermal:ti,ab OR Sebaceous:ti,ab OR Epidermoid:ti,ab OR Pilar:ti,ab) AND cyst*:ti,ab)) AND punch:ti,ab

MEDLINE (via PubMed) search strategy

#1 "Epidermal Cyst"[Mesh]

#2 Epidermal[tiab]

#3 Sebaceous[tiab]

#4 Epidermoid[tiab]

#5 Pilar[tiab]

#6 Cyst*[tiab]

#7 (#2 OR #3 OR #4 OR #5) AND #6

#8 #1 OR #7

#9 punch[tiab]

#10 #8 AND #9

EMBASE search strategy

S1 (EMB.EXACT.EXPLODE("epidermoid cyst"))

S2 ab(Epidermal OR Sebaceous OR Epidermoid OR Pilar) OR ti(Epidermal OR Sebaceous OR Epidermoid OR Pilar)

S3 ab(cyst OR cysts) OR ti(cyst OR cysts)

S4 S3 AND S2

S5 S4 OR S1

S6 (ab(punch) OR ti(punch))

S7 S6 AND S5

## Appendix D. The Trial Registry Search Strategy

ICTRP search strategy

(((Epidermal OR Sebaceous OR Epidermoid OR Pilar) AND (cyst OR cysts)) AND punch)

ClinicalTrials.gov search strategy

Condition or disease: ((Epidermal OR Sebaceous OR Epidermoid OR Pilar) AND (cyst OR cysts))

Intervention: punch

## Appendix E. Quality Scores for the Eligibility Studies for Others than Recurrence Rate

**Table A1.** Quality scores for the eligibility studies for mean operative time.

| Authors | Risk of Bias 2 Tool Assessment | | | | | |
|---|---|---|---|---|---|---|
| [Ref Number] | Bias Arising from the Randomization Process | Bias Due to Deviations from Intended Interventions | Bias Due to Missing Outcome Data | Bias in Measurement of the Outcome | Bias in Selection of the Reported Results | Overall Risk of Bias |
| [7] | Some concerns | Low | Low | Low | Some concerns | Some concerns |
| [4] | High | Low | Low | Low | Some concerns | High |

**Table A2.** Quality scores for the eligibility studies for mean length of postoperative wound.

| Authors | Risk of Bias 2 Tool Assessment | | | | | |
|---|---|---|---|---|---|---|
| [Ref Number] | Bias Arising from the Randomization Process | Bias Due to Deviations from Intended Interventions | Bias Due to Missing Outcome Data | Bias in Measurement of the Outcome | Bias in Selection of the Reported Results | Overall Risk of Bias |
| [7] | Some concerns | Low | Low | Low | Some concerns | Some concerns |
| [4] | High | Low | Low | Low | Some concerns | High |

**Table A3.** Quality scores for the eligibility studies for all adverse events.

| Authors | Risk of Bias 2 Tool Assessment | | | | | |
|---|---|---|---|---|---|---|
| [Ref Number] | Bias Arising from the Randomization Process | Bias Due to Deviations from Intended Interventions | Bias Due to Missing Outcome Data | Bias in Measurement of the Outcome | Bias in Selection of the Reported Results | Overall Risk of Bias |
| [7] | Some concerns | Low | Low | Low | Some concerns | Some concerns |
| [4] | High | Low | Low | Low | Some concerns | High |

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
