# Peer review of "Punch Incision versus Elliptical Excision for Epidermal Inclusion Cysts: Systematic Review and Meta-Analysis"

_2673-4095, doi:10.3390/surgeries2030033_

Round 1
Reviewer 1 Report
I read with interest authors' work. the described procedure is not of interest: I think that the cases analized are too few to draw any conclusion. also statistical analisis seems to be not so approprite
Reviewer 2 Report
Dear Authors,
It would be good if possible to further investigate the efficacy of the technique by studying each district in which the procedure has been applied. We may find that the technique works better in one part of the body than another or a specific type of cyst. Perhaps this could also be the start of another project.
Reviewer 3 Report
Interesting study regarding treatment of epidermal cysts. General surgeons, dermatologists and family physicians may benefit from this paper. I thetefore recommend its acceptance by the paper.
Kind regards!
Reviewer 4 Report
In this systematic review and meta analysis study entitled " Punch incision versus elliptical excision for epidermal inclusion cysts: Systematic review and meta-analysis", the authors successfully identified and reviewed the published outcomes of two different treatment approaches to the epidermal incision cysts (EIC). While this study has a sound design and methodology; the major limitation of it is that there were only 2 studies included in which moderate heterogeneity was present among them. However, this was due to the fact that the literature is very limited with regards to the current study's subject. Therefore, as the authors underlined, prospective randomized studies with large cohorts are still required to allow for better analyze the differences between the two approaches.
Round 2
Reviewer 1 Report
The study remains of low scientific inter est, with not significant statistical analysis
Reviewer 2 Report
Dear Authors,
thank you for the re-submission.
The paper provides information on this topic.
Reviewer 4 Report
Thank you for revising the manuscript in accordance with the reviewers' comments.